# Temporal variability of tidal and gravity waves during a record long 10 day continuous lidar sounding

Kathrin Baumgarten[1], Michael Gerding[1], Gerd Baumgarten[1], and Franz-Josef Lübken[1]

[1]Leibniz-Institute of Atmospheric Physics at the University of Rostock, Kühlungsborn, Germany

*Correspondence to:* Kathrin Baumgarten (k.baumgarten@iap-kborn.de)

**Abstract.**

Gravity waves (GW) as well as solar tides are a key driving mechanism for the circulation in the Earth's atmosphere. The propagation of gravity waves is strongly affected by tidal waves as they modulate the mean background wind field and vice versa, which is not yet fully understood and not implemented in many circulation models. The daylight capable Rayleigh-Mie-Raman (RMR) lidar at Kühlungsborn (54° N, 12° E) typically provides temperature data to investigate both wave phenomena during one full day or several consecutive days in the middle atmosphere between 30 and 75 km altitude. Outstanding weather conditions in May 2016 allowed for an unprecedented 10-day continuous lidar measurement which shows a large variability of gravity waves and tides on time scales of days. Using a 1-dimensional spectral filtering technique, gravity and tidal waves are separated according to their specific periods or vertical wavelengths, and their temporal evolution is studied. During the measurement a strong 24 h-wave occurs only between 40 and 60 km and vanishes after a few days. The disappearance is related to an enhancement of gravity waves with periods of 4-8 h. Wind data provided by ECMWF are used to analyze the meteorological situation at our site. The local wind structure changes during the observation period, which leads to different propagation conditions for gravity waves in the last days of the measurement and therefore a strong GW activity. The analysis indicates a further change in wave-wave interaction resulting in a minimum of the 24 h tide. The observed variability of tides and gravity waves on timescales of a few days clearly demonstrates the importance of continuous measurements with high temporal and spatial resolution to detect interaction phenomena, which can help to improve parametrization schemes of GW in general circulation models.

## 1 Introduction

The knowledge of atmospheric waves is crucial for our understanding of the circulation in the Earth's atmosphere. The propagation of different waves, e.g., gravity and tidal waves, and their interaction is a vital geophysical process, which couples the different atmospheric layers due to the transport of momentum and energy. Gravity waves (GW) and thermal tides differ in their sources. Gravity waves are mostly generated in the troposphere/lower stratosphere through the flow above orographic structures, convective instabilities, wind shears, jet streams or wave-wave interactions (e.g., Fritts and Alexander, 2003). Thermal tides are typically excited by solar heating of water vapor in the troposphere, ozone in the stratosphere and mesopause region as well as oxygen above 90 km altitude or also through latent heat release due to deep convection (Chapman and Lin-

dzen, 1970; Forbes, 1984; Hagan and Forbes, 2002). Due to the excitation process tides have periods of one solar day (24 h) and its harmonics like 12 h or 8 h. The tidal propagation can be either Sun-synchronous or not, and accordingly tides are called migrating or non-migrating tides (Forbes, 1995). They modulate the background wind field together with planetary waves and therefore have an impact on the propagation conditions for gravity waves (e.g., Eckermann and Marks, 1996; Senf and Achatz,

2011; Yiğit and Medvedev, 2017). Upward propagating GW transport energy and momentum and deposit them during their breaking and filtering to the mean background flow (Holton and Alexander, 2000; Fritts and Alexander, 2003). Models typically use only simplified linear parametrization schemes of gravity wave drag resulting in larger discrepancies between model and measurement data (e.g., Kim et al., 2003). Therefore, additional data are required for validation and more observational data are necessary for improving these parametrizations (Geller et al., 2013). There are approaches of gravity wave parametri-

zation schemes, which improve the structure and magnitude of tides, but a validation with observational data is still rare (Yiğit et al., 2008; Yiğit and Medvedev, 2017).

The middle atmosphere is one of the key regions for the interaction of gravity waves and tides. To investigate both wave phenomena different satellite, in situ (radiosondes and rocket soundings) and ground-based techniques (lidar, radar, and airglow measurements) were developed within the last decades (e.g., Gille et al., 2008; Hertzog et al., 2008; Preusse et al., 2008). Sa-

tellite data give an global overview of GW and tides. For instance, the climatology of tides in the MLT region has been revealed by temperature/wind observations such as the High Resolution Doppler Interferometer (HRDI), Wind Imaging Interferometer (WINDII) and the Microwave Limb Sounder (MLS) on board of the UARS satellite, or the TIMED Doppler Interferometer (TIDI) and the Sounding of the Atmosphere using Broadband Emission Radiometry (SABER) instrument on board of the TIMED satellite (e.g., Sakazaki et al., 2012). However, satellites typically need a large time interval of typically several weeks

to cover 24 h of local time. Consequently, any short-term variability in the dynamic features gets lost. Nevertheless, there are a few approaches to extract the short-term variability of non-migrating tidal modes from satellite data using a deconvolution method (Oberheide et al., 2002; Lieberman et al., 2015; Pedatella et al., 2016). But these approaches are limited to lower latitudes (<50° N) to resolve non-migrating tides (Oberheide et al., 2002). Therefore, this method is not suitable to resolve a day-to-day variability of tides at our latitudes. Radar measurements of horizontal winds produce nearly continuous data sets, from which

the short-term variability of gravity and tidal waves can be investigated, but only in a limited altitude range of approximately 70-100 km (Hoffmann et al., 2010). To cover the entire middle atmosphere the combination of different lidars using several scattering mechanisms (e.g. Rayleigh and resonance scattering) is the only measurement technique which provides temperature data from the troposphere/lower stratosphere to the mesopause region or even higher with a suitable temporal and vertical resolution to resolve the short-term variability. Lidar data provide vertical information of the atmospheric parameters over time

at the particular location. The Rayleigh-Mie-Raman (RMR) lidar located at Kühlungsborn is able to provide these information up to 75 km altitude without an additional resonance lidar. The advantage of the RMR lidar at Kühlungsborn is the ability to measure under nighttime as well as under daytime conditions resulting in a continuous temperature time series over the whole day (Gerding et al., 2013, 2015, 2016; Kopp et al., 2015). The most other lidar instruments which can measure during the day cover only a small altitude range (Chu et al., 2011), while other RMR lidars measure only during nighttime conditions

(Gardner and Voelz, 1987; Wilson et al., 1991).

This paper presents main features of wave activity at mid-latitudes for an altitude range from the lower stratosphere to the upper mesosphere on short time scales of 10 days in May 2016. To our knowledge, this is the longest continuous data set retrieved by a RMR lidar. The daylight capability of the Kühlungsborn RMR lidar as well as exceptionally good weather conditions make it possible to investigate wave structures over this time period, which allows to study the short-term variability of gravity waves and tides. The lidar data are analyzed in spatial domain on the one hand and in time domain on the other hand to distinguish between different waves either because of their vertical wavelengths or their periods. Data from the European Centre for Medium-Range Weather Forecasts (ECMWF) are used to characterize the background conditions in the troposphere and stratosphere based on hourly high-resolution forecasts (cycle 41r2 TCO1279/O1280). The organization of this paper is as follows. In Section 2 we describe our lidar instrument and how the data are treated. Section 3 presents the available temperature data during the 10 days of continuous lidar data in May 2016 and their related temperature deviations. In Section 4 we present the short-term variation of the gravity wave activity as well as the tidal activity. In addition to the lidar data, ECMWF data are used in Section 5 to characterize the background state of the atmosphere. The results are discussed in Section 6. Finally, the results are summarized and a conclusion is given in Section 7.

## 2 Instrumental Setup and Data

The Rayleigh-Mie-Raman lidar at Kühlungsborn was developed in 2009/2010 (Gerding et al., 2016). The transmitter mainly consists of a flashlamp pumped, injection-seeded Nd:YAG laser. We use the second harmonic of the laser output at 532 nm as emission wavelength due to a better signal-to-noise ratio compared to the fundamental laser output at 1064 nm. To measure during daytime, special spatial and spectral filtering techniques are used to suppress the solar background during the day. As a prerequisite for these techniques, the seeder is locked to an iodine absorption line for achieving high frequency stability and the laser beam divergence is reduced to ∼50 μrad using a 10x beam widening telescope. Afterward, the beam is guided co-axially with the receiving telescope into the atmosphere. The field of view (FOV) of the receiver is limited by a fiber cable with a small core diameter of 0.2 mm, resulting in a small field of view of only 62 μrad. The advantage is a reduction of the scattered background light from the Sun. A narrow band interference filter (IF) as well as two Fabry-Pérot-etalons (FPE) are used for spectral filtering. The IF has a full-width-at-half-maximum (FWHM) of about 130 pm. The etalons have a free spectral range (FSR) of about 120 pm (140 pm) and a FWHM of about 4 pm (4.5 pm), respectively. The FWHM of the etalons is on the order of the Doppler-width of the backscattered Rayleigh signal. This means that a small part of the backscattered Rayleigh signal is blocked depending on the actual Doppler-width of the backscattered light, thus, it depends on the atmospheric temperature at the particular scattering altitude. The reduced signal is not proportional to the atmospheric density anymore and therefore the classical retrieval for temperature is not valid. To overcome this issue, an altitude dependent transmission correction is applied for calculating absolute temperatures. Further information about the correction scheme and the validation can be found in Gerding et al. (2016).

To reduce the effects of tropospheric turbulence on the laser beam propagation and the alignment of laser and telescope FOV, an active beam-stabilization based on a Piezo-coupled mirror is used (Eixmann et al., 2015). Absolute temperatures

are retrieved by integration of the range corrected backscattered signal assuming hydrostatic equilibrium (Hauchecorne and Chanin, 1980). The initial temperature value for integration is taken from CIRA-86 (Fleming et al., 1990) in an altitude range between 70 and 75 km for the whole day due to the strong solar background at the Sun's maximum. The temperatures become independent from the start temperature approximately one scale height below the initial retrieval altitude. The integration time to retrieve the temperatures is 2 h with a temporal shift of 15 min. The vertical resolution is 1 km. Due to additional aerosol scattering below 30 km only temperatures above this altitude are taken into account in this paper.

To investigate different waves, temperature deviations from a slowly varying background field are determined. These deviations are retrieved by a subtraction of a mean temperature and by filtering in spatial and temporal domain using a Butterworth filter of fifth order. The filtering allows to distinguish between different wave components according to their specific vertical wavelengths and their periods. The cut off parameter are chosen as 15 km and 8 h for the vertical and the temporal filtering, respectively. A further description of the method can be found in Baumgarten et al. (2017). In addition, a composite analysis of the lidar data as described in Kopp et al. (2015) is used to investigate mean amplitudes of tidal waves.

## 3 Temperatures and temperature deviations

Temperatures from 4 May 07:45 UT til 13 May 2016 23:45 UT are shown in Figure 1 for an altitude range of 30 to 70 km. There are two small data gaps with a duration of ∼5 h in the beginning of 10 May and ∼1 h in the morning of 11 May due to weak cloud coverage. The highest temperatures of up to 280 K occur in the stratopause region (∼50 km). There is a large variation present around the stratopause region which is due to atmospheric waves. These variations are mainly caused by tidal waves as the observed periods are close to one solar day (24 h). This feature is weaker at the end of the 10 day period. To highlight the wave structures, temperature deviations from a mean temperature profile over the entire days are calculated and shown in Figure 1 as well. The overall variation seems to be dominated by a modulation of several days presumably caused by a planetary wave resulting in increasing temperatures with time below 40 km and decreasing temperatures above. Furthermore, this large scale variation is superimposed by dominant waves with periods of 24 h (e.g. 5 - 8 May) as well as various other waves. In general, these exceptional long data set does not only contain tidal waves, but also gravity waves as well as large scale waves presumably caused by planetary waves with periods of several days are visible.

To resolve the range of periods, which are occurring, the power spectral density is calculated from the temperature deviations from the mean temperature as a Lomb-Scargle periodogram for one particular altitude and smoothed with a Hanning window. The result is shown in Figure 2 in terms of frequency with an additional period scale for the altitude of 50 km. As already seen from the temperature deviations from a mean profile, different waves superimpose in the time series above our site. During the measurement a 24 h wave component is dominating the temperature deviations, but also waves with smaller periods of 5, 8, 12 h can be seen. For comparison, the expected slope of gravity waves of $-\frac{5}{3}$ is shown, which was found to be universal (VanZandt, 1982). In addition to gravity and tidal waves, also waves on planetary scales with periods of 48 h and larger than 100 h are observed. The error is estimated from the variability of three different spectra over the whole day, the first and the last five days of the observation. This is sufficient to show the potential variability of the spectrum.

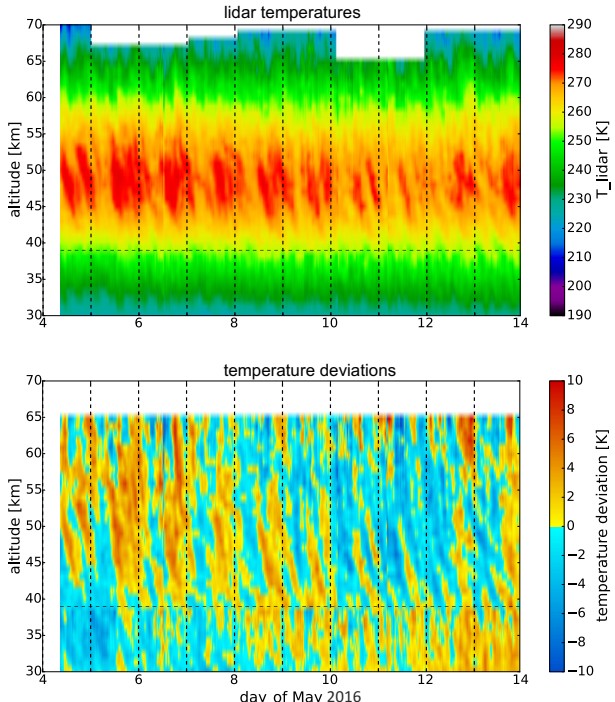

**Figure 1.** Temperature observed by the RMR lidar at Kühlungsborn on 4 - 13 May 2016 (at the top) and the temperature deviations from a mean temperature (at the bottom). The horizontal dashed line indicates the transition height between different receiver channels.

To analyze the variability of the waves further, a separation between different wave types has to be made. A 1-dimensional Butterworth filter of the fifth order is applied to extract temperature deviations induced by gravity waves. The cut off wavelength/period is 15 km/8 h, respectively, due to the assumption that tides have larger vertical wavelengths and periods. The resulting temperature deviations are shown in Figure 3 in panel a) for vertically and temporally filtered data. While the verti-
5    cal filtering (upper left panel) leads to wave structures with relatively small vertical wavelengths ($\lambda_z < 15$ km), the temporal filtering method (lower left panel) extracts only waves with periods smaller than 8 h and typically larger vertical wavelengths. The direct comparison of these two data sets shows differences in the regularity of the wave structures over the whole altitude range. Throughout the measurement time the gravity wave structures in the temporally filtered data (lower left panel) seem much more coherent than those of the vertically filtered data (upper left panel). Especially in an altitude range below 40 km
10    and above 55 km the structures from the vertically filtered data look less coherent what is related to a larger variation of frequencies. Within this altitude range (between 40 and 55 km) clear waves can be identified in most of the time. In the last few days the amplitude of the temperature deviations is increasing, especially on the 10-11 May. This indicates either different propagation conditions or different sources for these waves. That will be further investigated in the next section. To calculate the temperature variations that are induced by tides, the Butterworth filter is used as a low pass filter with the same cut off
15    parameters as for the gravity waves. These temperature deviations are shown in Figure 3 in panel b) for the vertical and the

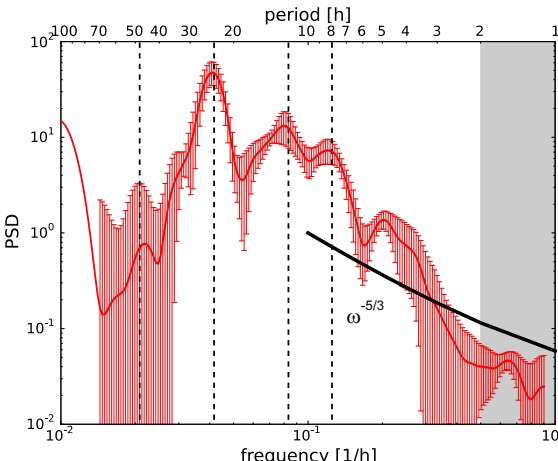

**Figure 2.** Smoothed power spectral density (PSD) as a function of frequency on 4 - 13 May 2016 calculated from the temperature deviations from a mean at 50 km altitude. The vertical dashed lines indicate periods of 8, 12, 24, and 48 h. The error bars denote the standard deviation of the power spectral density calculated from spectra obtained during the first and the second half of the measurement and the spectra for the whole measurement. The grey box marks the region where the data contain less information due to the resolution limit given by the integration time of 2 h.

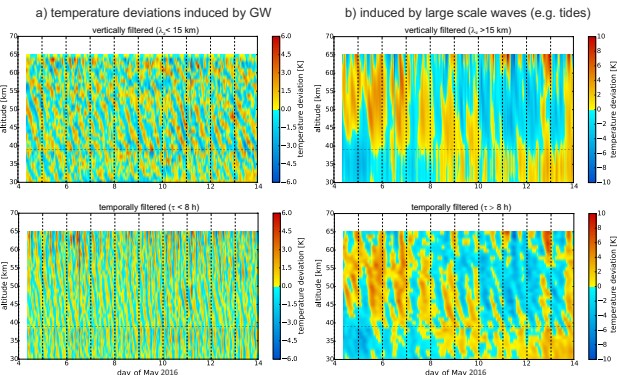

**Figure 3.** Temperature deviations for the vertically filtered (top row) and the temporally filtered data (bottom row) on 4 - 13 May 2016 induced by a) gravity waves and b) large scale waves (e.g. tides). The horizontal dashed line indicates the transition height between different receiver channels.

temporal filtering. The variations are dominated by large scale waves, which are later referred as tides. But we have to mention a localized ground-based measurement cannot provide information to distinguish between migrating and non-migration tides. Therefore, the tidal signatures measured by a lidar are not necessarily related to global tidal modes of the atmosphere. The variability of the different waves is shown in the next section.

## 4 Tidal and gravity wave variability

The variability of the waves observed is quantitatively investigated by using a wavelet transformation to calculate the periods of these longer and shorter scale waves. This is done by applying a Morlet wavelet of the fifth order to the filtered temperature deviations for a specific altitude. The temporal evolution of the periods is separately calculated for tides and gravity waves and is shown in the next two subsections.

### 4.1 Tidal variability

The wavelet spectra in three different altitudes for the so-called tidal observations are shown in Figure 4 for the vertically (left panel) and for the temporally filtered data (right panel). In the beginning and in the end of the measurement the wave amplitudes are over- or underestimated due to edge effects and these amplitudes are therefore not taken into account. The boundary to this so-called cone of influence is denoted by the white curved line.

At the lowest altitude of 40 km (shown in Fig. 4 in the lower panel) the vertically filtered data contain waves with a broad range of periods but with only small amplitudes of about 1 K. While the wave with a period of 24 h is visible over the whole sounding, other wave components occur more sporadically. The vertically filtered data show that there are less waves with periods between 8 and 14 h and vertical wavelengths of more than 15 km, while there is already a higher activity of waves with periods from 8 to 14 h from the temporally filtered data. This indicates that especially the semidiurnal and the terdiurnal wave components are differently represented in the vertically and in the temporally filtered data. As a result this means either these tidal components have smaller vertical wavelengths as assumed or the most occurring waves in this period range are related to gravity waves.

At 50 km altitude (shown in the middle panel of Fig. 4) the wave activity of especially the diurnal component is increased in the vertically filtered data as well as in the temporally filtered data. The first one shows amplitudes of 4 K for this component with the strongest occurrence on the 6 - 7 May 2016. Later this component becomes weaker. This behavior is even more pronounced in the temporally filtered data, where amplitudes of up to 6 K arise for the diurnal component in the first days. The amplitudes decrease to less than 1.5 K between the 10 May and the end of the measurement period. Other components with periods between 8 and 12 - 13 h are also visible, but they reveal smaller amplitudes and are less persistent compared to the diurnal component. The decrease over time of the diurnal component shown above indicates a strong short-term variability for tidal components, which has to be acknowledged. At an altitude of 60 km (upper panel in Fig. 4) this intermittency of the tidal signature becomes even stronger. The diurnal component completely vanishes after the 9 May 2016 for the vertically filtered as well as for the temporally filtered data and shows again a slight increase after the 12 May.

Especially the temporally filtered data do not solely contain tidal wave structures, instead there are also other longer periodic gravity waves included. To be sure that the potential bias caused by gravity waves is small, we also calculated the mean tidal amplitudes for the diurnal, semi- and terdiurnal component over three time intervals within May 2016. The number of days included in these intervals is given in Table 1. The calculation is based on the overlaying of temperature data for each of the days within the selected interval (Kopp et al., 2015). This composite of data is fitted with an harmonic function of fixed periods

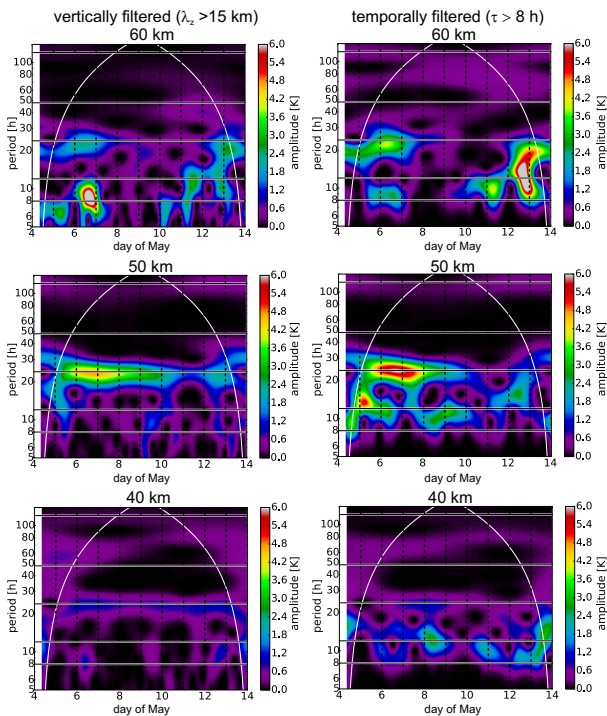

**Figure 4.** Wavelet spectra for periods of larger scale waves calculated from vertically (left) and temporally filtered data (right) on 4 - 13 May 2016 at an altitude of 40 km (at the bottom), 50 km (in the middle) and 60 km (at the top). The horizontal lines indicates periods with 8, 12, 24, 48 and 120 h. The white curved line shows the edge of the cone of influence.

**Table 1.** Time intervals in May 2016 for the composite analysis and the related number of days and measurement hours during each interval.

| Interval | no. of days | duration [h] | representation in Figure 5 |
|---|---|---|---|
| 01-09 May | 7 | 161.3 | dashed line |
| 10-28 May | 7 | 113.4 | dotted line |
| 01-28 May | 14 | 274.7 | solid line |

according to the solar tides. The mean amplitudes of the tidal components in each interval are shown in Figure 5 in comparison to data of the month May from former years (shaded area). The monthly mean amplitude in May 2016 (solid line) of the semi- and terdiurnal tide does not show a noticeable increase with altitude between 30 and 70 km. Only the diurnal component shows an increase of the amplitude up to 2 K in the altitude range of 30 to 50 km. Above, the amplitude decreases again and reaches a

5   value of 1 K at 60 km altitude. In comparison to this, the amplitude of the semidiurnal component varies only between 0.5 and 1 K. The amplitudes of the terdiurnal component are smaller than 0.5 K and they are therefore negligible compared to gravity waves with much larger amplitudes. Compared to former years only differences in the diurnal and semidiurnal component

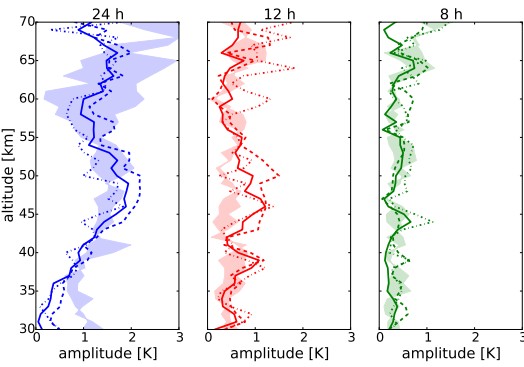

**Figure 5.** Amplitudes for the diurnal (left), the semidiurnal (in the middle) and the terdiurnal (right) component derived from a composite analysis during 14 measurement days in May 2016 (solid line), the first days of May (dashed line) and the last days of May (dotted line). A definition of the time intervals is given in Table 1. The shaded area shows the amplitudes derived from data in May 2012 and May 2014.

are visible. Especially around the stratopause the data of May 2016 show enhanced amplitudes for these both components in comparison to the data of May 2012 and 2014. The diurnal amplitudes show further differences below 35 km and above 65 km. This is probably due to phase differences during one particular month and different signal-to-noise ratios during the years.

Looking at the single time intervals reveals huge differences in the amplitudes of the diurnal component over the month,
especially above 43 km altitude. The small variability below this altitude indicates a constant excitation over the whole time period, otherwise the differences mentioned before would also occur in this altitude range. The results from the wavelet analysis for the diurnal wave component at an altitude of 40 km also supports this statement (see Figure 4 at the bottom). Above 43 km the amplitude of the diurnal component is significantly larger in the first interval as in the monthly mean. This is not visible for the other components. For those the amplitudes are partly larger in the first time interval, but at other altitudes they are larger
in the second time interval. On the contrary, the amplitude of the diurnal component is constantly smaller in the last days of May than in the mean. The differences of the amplitudes during the different intervals of minimal and maximal amplitudes are about 30-50% of the absolute value. The results for the diurnal component are in agreement with the temporal evolution of this component in the wavelet analysis, aside from the slightly different amplitudes. The origin of these differences in the amplitudes lies in the filtering methods, which are not exclusively restricted to tides, also other long periodic waves are included
in the data. Additionally to this, some long periodic gravity waves which could be Doppler shifted to observed periods larger than the Coriolis period may also be included. These waves would lead to higher amplitudes in the results from the wavelet analysis compared to those from the composite analysis. The composite analysis is based on the assumption that tides have constant phases, while gravity waves have randomly distributed phases due to their different sources. Consequently, gravity waves are averaged out during the analysis even if they have periods similar to tides. The calculated phases of the diurnal
tidal component stay constant in the different time intervals in the altitude range above 40 km (not shown here). This leads to the assumption that the dominating tidal Hough modes have not changed during the measurement period. To summarize this,

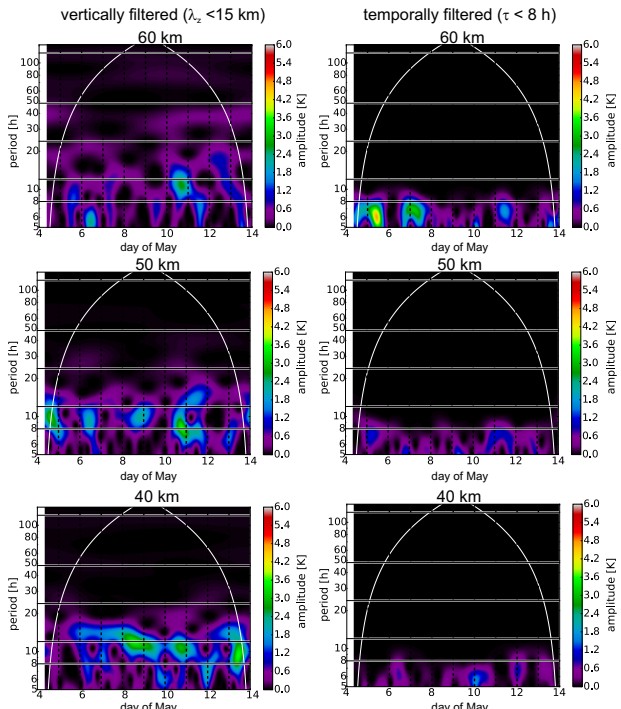

**Figure 6.** Wavelet spectra for periods of gravity waves calculated from vertically (left) and temporally filtered data (right) on 4 - 13 May 2016 at an altitude of 40 (at the bottom), 50 (in the middle) and 60 km (at the top). The horizontal lines indicates periods with 8, 12, 24, 48 and 120 h. The white curved line shows the edge of the cone of influence.

the amplitudes of tides vary with time, but differently for different altitudes. They may increase at a certain altitude without a corresponding enhancement at other altitudes.

## 4.2 Gravity wave variability

As a comparison the same analysis is done for temperature deviations induced by gravity waves. The calculated wavelet spectra
5 for the same altitudes are shown in Figure 6 for the vertically (left panel) and the temporally filtered data (right panel). The periods look quite different than in the spectra from the tidal wave features shown in Figure 4. The dominant periods are much smaller than 24 h in most of the time and at every altitude between 40 and 60 km, even if the filtering was done with respect to vertical wavelengths. For the temporally filtered data this is per construction due to the cut off period.

At 40 km altitude waves with periods between 6 and 12 h are observed from the vertically filtered data with amplitudes of up
10 to 3 K on several days during the sounding. The largest amplitudes are observed for waves with periods of about 10-12 h (see Fig. 6 in the lower left panel). The temporally filtered data reveal lower amplitudes of about 0.5-1.5 K of waves with periods smaller than 8 h (see Fig. 6 in the lower right panel).

At an altitude of 50 km (shown in the middle panel of Fig. 6) the wavelet amplitudes are slightly reduced and large amplitudes occur more sporadically. On 10/11 May a wave with a period of 8 h becomes strong for the vertically filtered data. The appearance of this wave is even more pronounced at 45 km altitude (later shown in section 5). The temporally filtered data also show such a wave signature, but with a reduced amplitude due to the cut off period of 8 h. The GW signatures from the vertically filtered data are less pronounced above an altitude of 60 km (Fig. 6 upper left panel), except for the signature on the 10 May.

As seen in Figure 4 at 50 km altitude, a strong diurnal component is visible for the tidal wave features. This feature decreases at the same time when the gravity waves with a period around 8 h become important. Both phenomena are reduced above the stratopause. Therefore, we assume that there is a close connection between these two kinds of waves due to a possible wave-wave or wave-mean flow interaction. To study this further, it is necessary to investigate the mean background state of the atmosphere. This is done in the next Section with the use of ECMWF data of the integrated forecast system (cycle 41r2).

## 5 Meteorological situation

The propagation of tidal and gravity waves depends on the mean background wind as well as on the interaction of tides and gravity waves. A change in the excitation of tidal waves could also lead to temporal differences in these waves (e.g., Achatz et al., 2008). ECMWF data provide temperature, ozone and horizontal wind information. The data above Kühlungsborn are studied to reveal if there were changes during the sounding period. The temperature data provided by ECMWF is shown in Figure 7 to make sure that ECMWF is able to reproduce the meteorological situation above Kühlungsborn. Therefore, ECMWF temperatures and the temperature deviations from a mean temperature profile are shown for the same altitude range as the lidar data.

In general, the comparison to the lidar data (Fig. 1) shows similar structures. ECMWF temperature deviations from a mean temperature reveal also strong wave structures with periods of 12 and 24 h. Especially in the altitude range between 40 and 50 km the phases of the wave structures are very similar to the lidar temperature deviations (see Fig. 1 at the lower panel). But for higher altitudes the similarity gets lost as ECMWF shows a wave structure with a much longer period compared to the lidar data. This discrepancy is probably related to the sponge layer of ECMWF at 50 km and that there is no data assimilation anymore (e.g., Jablonowski and Williamson, 2011). In the altitude range between 30 and 40 km smaller differences between the lidar and the ECMWF data are present, e.g., the tilt of the phase lines differs among these two data sets. But the overall wave structures, especially for tidal waves with periods of 12 h and 24 h, are similar up to an altitude of 50 km, while shorter periodic wave structures are different in both data sets. However, ECMWF data provide useful information at least of the background atmosphere and are suitable to get a comprehensive understanding of the state of the atmosphere.

The overall zonal and meridional wind above Kühlungsborn derived from ECMWF is studied to reveal if there are changes for the propagation conditions of the waves. Therefore, the wind data is filtered using a low pass Butterworth in time with a cut off period of 30 h to get only the background wind without changes due to gravity or tidal waves. The wind structure is shown in Figure 8 in an altitude range from the ground to 60 km. As ECMWF data have a sponge layer at ∼50 km and there

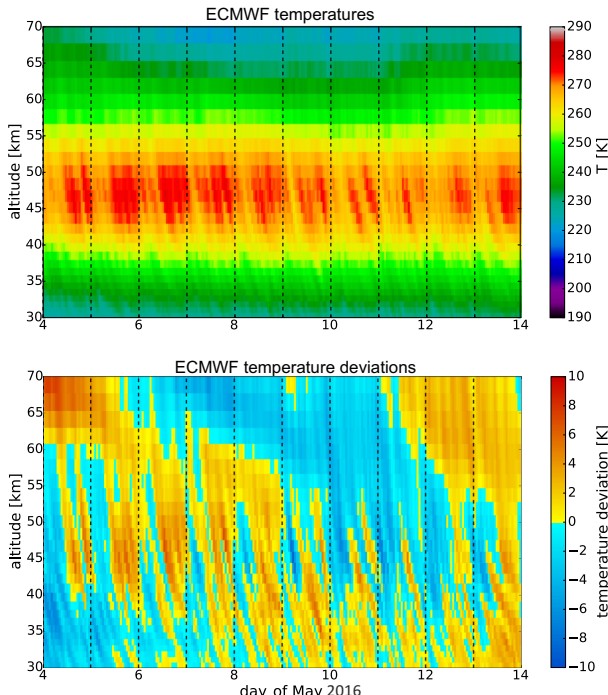

**Figure 7.** Temperature provided by the integrated forecast system of ECMWF (cycle 41r2) extracted with 1 hour temporal resolution on 4 - 13 May 2016 (at the top) and the calculated temperature deviations from the mean temperature (at the bottom).

are basically no data above the stratopause assimilated, the reliability of the data decreases above this altitude. In general, the zonal and meridional winds show large temporal variations mainly in the upper troposphere.

   While zonal winds at the altitude of about 10 km are weak and towards the East in the first days, the wind veers to the West on the 7 May for about 3 days. After this time an even stronger zonal wind towards the East is reestablished with wind velocities 5 up to 18 m/s. At the same time a wind reversal in the meridional wind occurs. This wind component is blowing to the South in the first days. On 7 May the wind veers to the North until the end of 9 May. The wind reversal in the zonal and meridional component at the end of 9 May coincides with an increase of gravity wave activity and a disappearance of the diurnal tidal component after this point in time. We will examine the relation between gravity waves and the background wind further in Section 6.

10   Above 15 km altitude the zonal wind is generally weak. Between 15 and 20 km the wind direction is mostly towards the East in the last days. Above this altitude range the wind is blowing most of the time towards the West. At an altitude between 40 and 50 km the wind is slowly decreasing from eastward wind to weak westward wind. However, the overall variation of the horizontal wind above the tropopause is presumably caused by planetary waves. As the planetary wave activity is in general week during summer, the overall wind variation is weak as well.

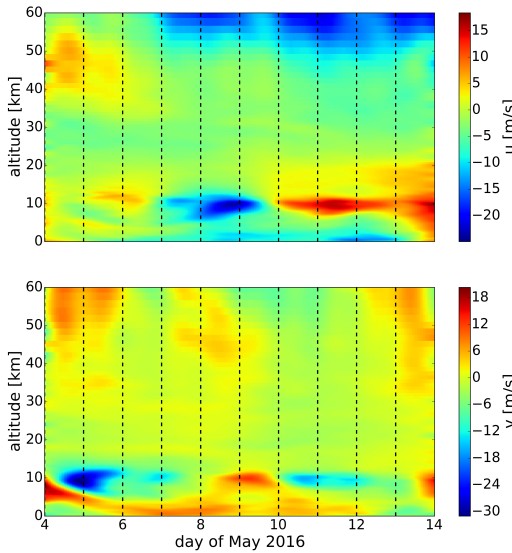

**Figure 8.** Zonal and meridional wind above Kühlungsborn on 4 - 13 May 2016 derived from ECMWF. The data has been low pass filtered with a cut off period of 30 h.

We have also studied the ozone distribution provided by ECMWF as the excitation of solar tides is related to the absorption of solar radiation by ozone in the stratosphere in addition to water vapor in the troposphere. In Figure 9 the temporal evolution of the ozone concentration is shown up to an altitude of 60 km. The maximum of the ozone layer is located at an altitude of about 22 km with additional strong layers between 12 and 18 km. In the lower stratosphere the ozone shows a larger variability

5   with a maximum on the 9 May. Above the maximum of the ozone layer, the ozone concentration decreases rapidly. However, we found no correlation between the time interval of the increasing ozone and the occurrence of the diurnal component. A closer look on parts of the ozone layer reveals a similar behavior of the ozone in an altitude range of 30-40 km as the diurnal component at ∼50 km altitude in the lidar data, but the relative change of the ozone is only 2%. It seems to be unlikely that these small ozone variations are the reason for a changing excitation of the diurnal tide during the sounding period. This is

10  even more reliable, if we have a look on the much larger differences in the ozone below 30 km, which are clearly not correlated with the appearance of the diurnal tide. In general, a local change in the ozone layer is not expected as an relevant reason for a change in the global tidal wave field. The observed local changes in the diurnal wave signatures may be caused by variations of the ozone layer at other longitudes as tides are a global structure.

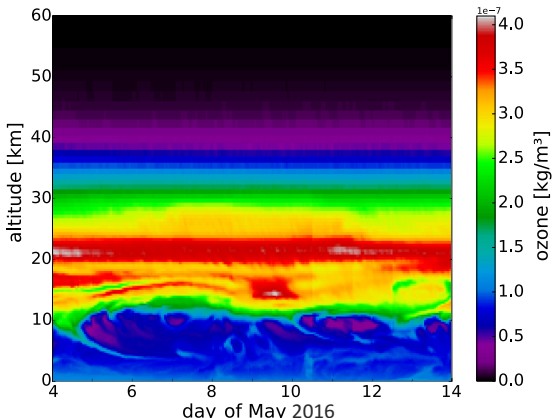

**Figure 9.** Ozone density from the ground to 60 km altitude above Kühlungsborn on 4 - 13 May 2016 derived from ECMWF

## 6  Discussion

The daylight capability of the RMR lidar at Kühlungsborn allows us to study temperatures in the middle atmosphere during night and day and evan for a few consecutive days depending on cloud free conditions. During May 2016 an exceptional measurement lasting 10 days was performed at our site. Most other multi-day lidar studies are done with resonance lidars in
an altitude range of 80-110 km because technologies for daylight suppression are available since many years for these type of lidars (e.g., States and Gardner, 2000; Fricke-Begemann and Höffner, 2005; Yuan et al., 2010; Cai et al., 2017). Another 3-day study for the middle atmosphere was performed by Baumgarten et al. (2015) also using an RMR lidar but at high latitudes. They investigated only inertia gravity waves in temperature data with combined wind measurements without looking on the tidal variations.

Our unprecedented measurement reveals a strong variation of different atmospheric waves, especially around the stratopause. Between an altitude of 40 and 50 km the diurnal tidal wave component shows an increase as expected due to the decreasing air density. But surprisingly this increase is not visible during the whole time. Above 50 km the diurnal tidal wave component strongly decreases again. A nearly identical behavior of this tidal component is revealed using a composite analysis as an independent calculation method to determine a mean tidal amplitude. The appearance of the diurnal wave component in a
particular altitude range is frequently related to a trapped mode of the tide, which can not propagate upward. Such a local maximum of the diurnal tide has also been reported by e.g. Forbes and Wu (2006) and Gan et al. (2014). However, results on the temporal variability are generally rare. To find possible reasons for the variability of the tidal component in our observations, ECMWF data from wind and ozone are investigated (shown in Figure 8 and 9). The ozone presumably leads to the excitation of the diurnal tide in the stratosphere as expected from theoretical studies (Forbes, 1995). But the localized ozone change is
not the reason for the disappearance of the diurnal tide in the last days of the sounding because of two indications. First, the ozone concentration shows only small variations of ∼2% over the time between 30 and 40 km altitude compared to the huge

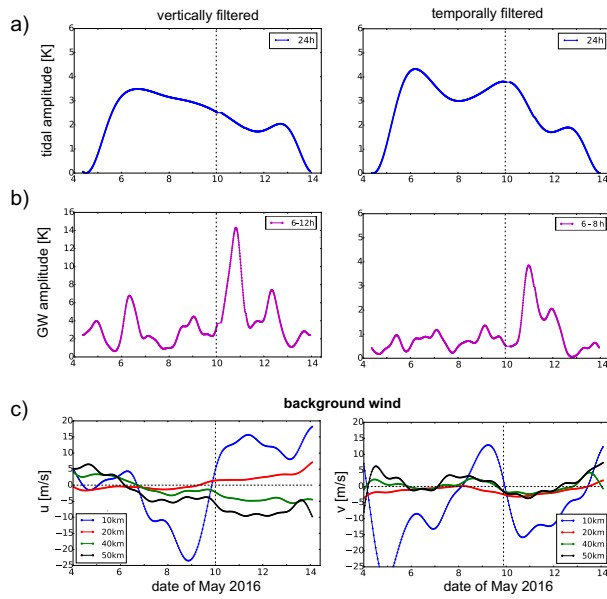

**Figure 10.** Wavelet amplitudes of a) the diurnal component and b) gravity waves at 45 km altitude compared to c) the zonal and meridional wind from ECMWF at different altitudes. The vertical dashed line indicates 0 UT on 10 May.

amplitude variation of the diurnal tide of ∼50%. And second, the amplitude of the diurnal component is constant over the time at 40 km altitude, which is related to a constant excitation of this tide. Nevertheless, the lack of correlation at a local site is not an evidence that they are not related because a weaker ozone layer at other longitudes causes also a weaker global diurnal tide. But if this would be the case, then the excitation of the diurnal tide would also not be constant at 40 km. However, we have not

5 observed this.

The strong correlation of the temporal evolution of the amplitudes of tidal and gravity waves and the winds suggests that there might be a close connection between each other. This topic is highlighted in Figure 10 for an altitude of 45 km. Atmospheric waves propagate within the mean flow. The consequence of this is a possible Doppler-shift of the real intrinsic period to an observed period. This period is observed by a ground-based instrument. To take this into account we summed up all

10 amplitudes from waves with periods between 6 and 12 h (6 and 8 h) for further analysis to form a group of gravity waves for the vertically (temporally) filtered data, respectively. This is done under the estimation that a shift of the period is small as the horizontal wind components illustrate only small changes above the tropopause with time. Unfortunately, we do not know the propagation direction of the waves, which would be necessary to calculate the intrinsic wave periods from the observed ones. The consequence of summing up waves is that the amplitudes shown in Figure 10 are overestimated. However, only the

15 temporal evolution is relevant here. As already seen in the wavelet analysis there is a strong diurnal component, which reaches an amplitude of up to 4 K on 6 May at 12 UT. Compared to the seasonal variation of this diurnal oscillation in the stratopause region these amplitudes are stronger than the mean values of May provided by Kopp et al. (2015). Usually, the expectation

is a reduced tidal amplitude during summer, which is not observed here. Later on in the study here, the amplitude is indeed slowly decreasing for the vertically as well as the temporally filtered data. The minimum of this wave component is reached on 11 May for both filtering methods and the amplitudes are reduced by a factor of 2. Compared to this, the amplitudes of gravity waves are strongly increased on 10/11 May. The absolute value of the amplitudes are overestimated due to the inte-

gration across a range of periods, but nevertheless the increase is significant at this time, when the diurnal wave component is weak. The amplitude of the zonal and meridional background wind is also shown at different altitudes in Figure 10 panel c). The weakening of the diurnal wave component does not coincide with the wind reversal in the tropopause region. But it is reliable that after this wind reversal the gravity waves become important, especially those with periods <8 h. This indicates a difference for either the vertical propagation conditions for these waves from the same source or a better oblique propagation

of gravity waves from different sources. The first indication exactly leads to a behavior, which was already observed in the seasonal variation of these certain waves during summer in the study by Baumgarten et al. (2017). In this time these waves can propagate to higher altitudes compared to inertia gravity waves. The direction of the mean winds below 30 km altitude is typically towards the East during summer. This leads to a filtering of most of the waves except these with a high phase velocity towards the East.

In the next part of the discussion we want to demonstrate that the decrease of the diurnal tidal wave component is correlated with an increase of the gravity waves from the temporally filtered data and not with inertia gravity waves. We estimate this because the gravity waves from the temporally filtered data appear only in a limited altitude range of 42 to 52 km just in the same altitude region where the diurnal tide decreases (see Figure 11). Contrary to this, the inertia gravity waves show also a strong occurrence in the first days of the measurement and at lower altitudes (partly visible in Figure 6 on the left hand

side), when also the diurnal tidal wave is strongly present. The strongest occurrence of gravity waves with periods <8 h is slightly before the diurnal tidal wave component reaches its minimum. To demonstrate that this is closely linked, the altitude dependencies of the amplitudes of the diurnal tide as well as of gravity waves from the temporally filtered data are shown in Figure 11 for two certain times, namely when the minimum of the tide is reached and when the maximum of the GW occurs. For this analysis both types of waves are differently treated. While the maximal amplitude of the GW along the altitude is

normalized to one, the minimal amplitude of the diurnal tide is also set in relation to the maximum of the tide on 6 May before a normalization along the altitude is done. This means, in each altitude the maximum of the tide is normalized to one, and the minimum values on the 11 May are stored. After this a normalization along the altitude is done as for the gravity waves. For the tidal amplitudes it has to be taken into account, that the amplitudes change over the whole time, therefore we have to relate the minimum to the overall temporal change of the tide. The result shows an occurrence of the gravity waves just between 42 and

52 km. Within this altitude range the amplitude of the diurnal tide starts to decrease. When the gravity waves disappear also the decrease of the amplitude of the diurnal tide stops. We speculate that our observations show an interaction between the gravity waves and the diurnal tide resulting in a suppression of the diurnal tide for several hours. Our speculation is compatible with investigations by Ribstein et al. (2015). They have demonstrated a strong impact of GW on a slowly varying background, as it is caused by tides, and vice versa using model data. Most other weather and climate model studies are using simplifications

(i.e., only vertical propagation) which lead to overestimated amplitudes of GW in the mesosphere and thermosphere, which

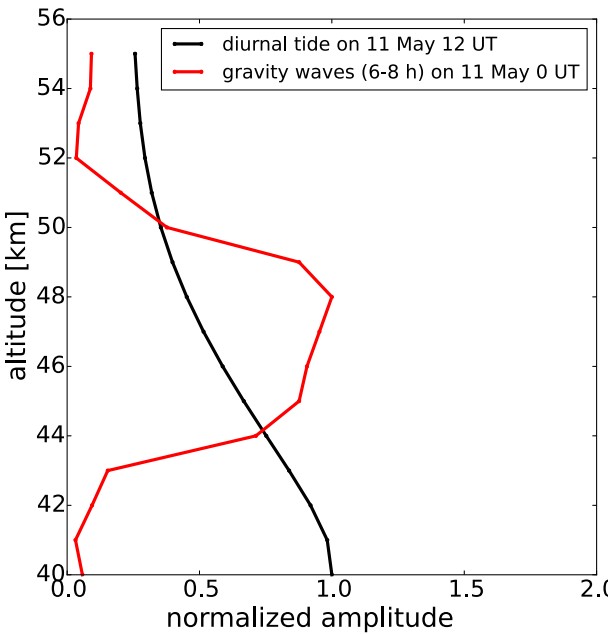

**Figure 11.** Normalized wavelet amplitudes of the diurnal wave component and of the gravity waves from the temporally filtered data on 11 May at 12 UT (0 UT) as a function of altitude.

also lead to discrepancies for tidal amplitudes (Alexander et al., 2010; Senf and Achatz, 2011). Solely from our observations we cannot decide which process results in the weaker diurnal oscillation. The interaction of gravity waves with the diurnal tide is only the simplest and most obvious one. From a lidar measurement no distinction between migrating and non-migrating tides can be obtained. Therefore, we cannot exclude an interaction of different tidal modes, which could also lead to a decrease
5   in the observed diurnal oscillation, although no changing phase was detected.

A closer look to the observations shown here reveals that the disappearance of the diurnal wave starts even before the gravity waves become prominent. We estimate that this is related to a stronger Doppler-shift of gravity waves to an observed period of 24 h in the first days of the sounding period. If these gravity waves have large vertical wavelengths and these large periods, they would contribute to the results for tides. This would explain that the composite analysis shows smaller amplitudes for the
10  tidal components than the wavelet analysis. Furthermore the wind shows changes from weak eastward wind to westward wind in this altitude range above 40 km, which could lead to the Doppler-shift mentioned above. But in the end, we cannot prove this because of missing modeling data with sufficient resolution and accuracy in this altitude range. Nevertheless, this study does indicate a wave-wave interaction of gravity waves and the diurnal tide.

## 7 Conclusions

In 2016 an unprecedented time series of temperature observations by lidar of about 10 days in the middle atmosphere at mid-latitudes showed a large temporal variability of local tidal waves. Especially the amplitudes of the diurnal wave decreased during the last half of the sounding period. This means tides are highly variable even on periods of a few days. This needs to be taken into account when the sampling of tides occurs rather sporadic, e.g. by satellites. Basically due to different wind conditions during the measurement time in the tropopause there was a change in the propagation conditions for gravity waves. We conclude that this leads to the observed wave-wave interaction mainly between tides and GW with periods smaller than 8 h and therefore a disappearance of the tidal component in the last days of the measurement in an altitude range of 42-52 km. Both types of waves decrease above the stratopause region, which could be related to a destructive interference of these waves in addition to a trapped mode of the tide. Such a behavior is not well reproduced in model studies which use simplified linear parametrization schemes of gravity waves. The results of the study shown here highlight the necessity for a more sophisticated parametrization of GW in climate models. Newly developed whole atmosphere parametrization schemes of gravity waves could provide new insights to understand tidal-gravity wave interaction (Yiğit et al., 2008). But this is out of the scope of this paper and will be done in a future study as this data set provides a possible benchmark for comparisons with models.

*Acknowledgements.* The data for this paper are available upon request. We gratefully acknowledge Maren Kopp for her help in the installation of the daylight capable RMR lidar as well as Josef Höffner for his contribution to the beam stabilization. We thank Michael Priester and Torsten Köpnick for the maintenance of the RMR lidar system at IAP. We also acknowledge all our students for their numerous hours of lidar operation. This project was partly supported by the Deutsche Forschungsgemeinschaft (DFG, German Research Foundation) under project SPP1788 (DynamicEarth) - CH1482/1-1 (DYNAMITE) and under project LU1174/8-1 (PACOG), FOR1898 (MS-GWaves). The work was also partly supported by the Bundesministerium für Bildung und Forschung (BMBF, Federal Ministry of Education and Research) under project D/553/67210010 (ROMIC-GWLcycle).

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
