# Peer review of "Temporal variability of tidal and gravity waves during a record long 10 day continuous lidar sounding"

_Atmospheric Chemistry and Physics, 2017_

## Referee Comment (RC1) · Anonymous Referee #1 · 30 Jul 2017

The paper by Kathrin Baumgarten and coauthors presents a continuous observation of middle atmosphere temperatures of unprecedented length. This is facilitated by newest advances in daylight capability as well as favorable observation conditions. The data are analyzed with a number of different techniques discussing the different effects of vertical and horizontal detrending. This is comprehensive and well presented. However, I do not agree with the interpretation of the findings in terms of a GW-induced change of the tidal amplitudes. This interpretation is based solely on the data without a clear discussion of the mechanisms. I would ask the authors to reconsider this part in the light of the major comments given below.

Major comments

The central figure in terms of interpretation is Figure 10. The figure presents the temporal variation of the tides, the GWs and the background winds. The tides show a general tendency of decrease over the 10-day period with some variation seen only in the temporally-filtered data. GWs show several peaks with the strongest peak after day 10, after the wind reversal of both zonal and meridional winds. That peak is more outstanding in the temporally filtered data, but amplitudes are much larger in the vertically filtered data.

The only point which supports your hypothesis is that the strongest decrease in tidal amplitude and the strongest peak of GW amplitudes appear both at the same day, i.e. day 11, in the temporally filtered data. However, this is a single event! It may be coincidence, the two variations could be caused by a common reason or it could indeed be causality. In order to argue that it is not a coincidence you would need more data. You claim that you can find a strong correlation: that could be tested by actually calculating correlation coefficients. (Proof me wrong, but I do not expect you to find a correlation significant based on Student's t-distribution.)

In order to argue for causality you would need to describe the mechanism in more detail. Here I see several difficulties.

A) The (part-)global nature of tides. In general tides are understood to be a superposition of Eigenmodes of the atmosphere resonantly excited by solar heating (or chemical heating or latent heat release triggered by the solar cycle). In the superposition of different modes there may be locally varying amplitudes depending on the place of the observation. The relative phases then also would decide on the amplitude at a single observing station. In addition, non migrating tides may be excited by the interaction of tides and planetary waves (cf. e.g. Liebermann et al., JGR-Atmos., 2015). By the way, it is not correct that satellites do not observe short term variability, see this paper and also e.g. Pedatella et al., JGR-space, 2016.

If you want to understand the cause for the variation at Kuehlungsborn you have to first find out which tidal modes are involved and whether these modes change amplitude or rather phase. Then you would need to look for mechanisms on a part-global scale. The column above Kuehlungsborn is definitely insufficient.

B) You would still have a stronger argument if you could show that the event considered is exceptionally strong. A single event may dominate the momentum flux of a whole latitude circle. You have climatologies from you own observations as well as from satellite data. Put your event into a context! That is a value of its own. In your discussion at the moment you focus on the temporally filtered data, because there you have only on peak while you have three in the vertically filtered data. However, GW momentum flux is proportional to the square of the temperature amplitude, which is a factor 9 (amplitude squared) larger in the vertically filtered data. Also the latter would be more susceptible to wind filtering and dissipation.

C) Also GWs propagate obliquely. And what causes the peaks? The question what the origin of the peaks in GW amplitude might be is probably the most obvious one. Again you try to answer this solely from the column above Kuehlungsborn. The wind reversal points to some cyclone or anticyclone in the vicinity. Is there frontal activity? Please consider also weather maps and search for potential sources. Also high resolution ECMWF data may be helpful (you show quite remarkable agreement) to find potential sources.

Overall, my recommendation is to strengthen the observational part by putting the observed event into a context: is it an unusually strong event? In addition, dig somewhat deeper into the sources of the GWs with the help of NWP data, potentially considering high resolution ECMWF data. Could UWaDi be helpful? Keep the ozone discussion short, local ozone does not tell you much. Then at the end you may suggest that the GWs could have an influence on the tides but do not build your whole paper around this hypothesis and do not quote this as an "elegant demonstration of the strong impact of gravity waves on the diurnal tide": you are missing the evidence!

Specific comments:

P1L2 infected -> affected

P1L22 Separate your references for GWs and tides. There is also tidal excitation due to latent heat release, i.e. convection.

P2L1 therefore -> accordingly

P2L10 ground based = what remains on the ground = radar, lidar, airglow radiosonde, rocket: in situ

P2L35 data during the 10 days of continuous lidar data in May 2016 and

P3L6 flash-lamp pumped

P3L7 better signal-to-noise? In which way?

P3L27 below the initial retrieval altitude ?

P4L9 over the entire period are shown in Figure 1 as well.

P4L12 only contain tidal waves*;* also gravity waves as

P5L1 calculated using as a <- one or the other

P5L2 Please include the 50km altitude also in the text

P5L17 Within this altitude range clear waves What do you mean? The altitude range 40-55km or the ranges below and above? I cannot follow your discussion here: In both cases I see waves, though at different frequencies. Maybe waves in the upper panel weaken at the highest altitude. Is that the effect of the vertical Butterworth filter? It also appears to me that the change between receiving channels also causes a non-continuity in the waves - though that is hard to say as the line is in.

P6F3 vertically-filtered data

Please show also the low-pass filtered data in Fig.3

P7F4 Wavelet spectra

P8L1 The unexpected decrease of the diurnal component shown above indicates a strong short-term variability for tidal components What precisely do you want to infer here? That the tides have a boost at 50km which not yet reached 60km? Then: what is the vertical group velocity of tides? Your period is quite long. Shouldn't the high amplitudes you see at 6 May reach a few days later also 60km? Or do you mean simply that there is a strong variation at 50km with a peak at 6 May? Please clarify.

P8L1 referred to be constant (e.g. from the satellite community). a) Also in the satellite community there are quite a number of approaches b) Often some temporal coherency is used to identify a number of different migrating and non-migrating modes. These, evaluated at a single location, may result in apparently strongly varying amplitudes, albeit a constant amplitude was assumed for all these global modes. c) That amplitudes are assumed to be constant is then usually just the lack of data. So far we unfortunately do not possess the perfect observations with simultaneously global coverage and good coverage of local time.

Please omit the half sentence here. You can include some discussion in the introduction or summary section.

P8L15 That you need a closer look tells you that the differences cannot be huge ... Are you even sure that they are significant? Probably yes, as they seem to be consistent over a wider altitude range. However, that they match at 43km does not seem to be significant to me given the variations in the profile above.

P8L22 50 % : Sorry, I am bewildered. In Fig. 5 left panel, blue curves you have at 44km a mean amplitude of roughly 1.5K, and a minimum (dashed) of 1K = 30%. Just above at 46km the three curves are inside .2K at 2K mean amplitude, i.e. about 5% variation around the mean. For a global mode with long vertical wavelength such as a tide I would not assume the minor zigzagging to be real but indicative of the precision with which you can determine your amplitudes. On the other hand, the wavelet analysis

indeed shows a change of a factor 2 in amplitude, so I am missing that special part of consistency. My interpretation would be that tidal amplitudes are stable, if periods of 7 days and larger are considered but that on shorter scales the local tidal amplitudes vary strongly.

P8L26 which could be Doppler shifted to *intrinsic* periods larger than the Coriolis period

P8L27 In the composites you implicitly assume a constant phase of the tide over the analysis interval. Phase variations hence would also be a reason for different results.

P9L17 we assume that the disappearance -> we want to investigate whether ????

P10L2 depends on wind conditions as well as on their interaction. Please be more precise, e.g. the propagation conditions of tides depend on the mean background winds and the propagation of GWs both on the mean wind and the tides.

P10L6 ECMWF is able to reproduce the meteorological situation above Kuehlungsborn.

P11L8 the sponge layer and the fact that there are basically no data above the stratopause assimilated.

P12L5 What do you mean: that the variation, albeit weak, is caused by PW or that the weakness of the perturbation is caused by PW

Fig10: Please assign panel indices (a,b,...). There seems to be a data gap after day 10 in the observations. There are some odd blue lines at the bottom of the plots in the middle row.

---

## Referee Comment (RC2) · Anonymous Referee #2 · 17 Aug 2017

This paper presents a study of gravity wave and tidal activities in the stratosphere and lower mesosphere using lidar temperature measurements that span continuously over 14 days. Such continuous measurement is unprecedented and is extremely valuable for the study of wave propagation through this region and interaction of waves at different time scales. This work is fairly thorough, and the key points are well described and supported by analyses from various angles. Because of this, I think this work should be published. I do find some various places where the manuscript can be improved and have given my suggestions below.

Aside from that, it's important to note that tides are global features while GWs are

[Figure]

highly localized. The decrease of diurnal oscillation seen in this measurement does not necessarily mean a decrease of the global diurnal tide amplitude. Such decrease could be due to interactions among different tidal modes or with planetary waves. Nevertheless, I think the analysis in this work does make a case that it is possible the local decrease of the diurnal oscillation is related to a GW. The authors do need to carefully distinguish between diurnal tides (global) and diurnal oscillation (local) in the text.

page 2, line 25: This is perhaps not a fair comparison. A lidar's capability is not measured by the altitude range it can measure (unless they all measure the same thing). The Rayleigh and metal lidars measure different regions and serve different purposes. If the lidar data used in this study included the mesopause region, then this statement would be appropriate.

page 3, line 26: It is not clear how 'strong solar background' is related to the starting altitude. Does it make it lower or higher? Please explain.

page 3, line 31-32: please specify the cut-off periods/vertical wavelength of these filters.

page 4, line 12: suggest changing to '... data set contains not only ..., but also ...'

page 5, line 1: remove 'using'

page 5, Figure 2: Since the data is averaged for 2 hrs (stated on page 3, line 28), the highest frequency that can be resolved is 1/(2hr). Even though the data point is every 15 min, the figure better not extends to higher frequency because there is really no information beyond 1/(2 hr). Caption: not sure what the 'first' and 'second' half mean and how they relate to the error bars. Need clarification. It's not clear what data was used to calculate this spectrum. Is it from 50 km temperature only or average over an altitude range around 50 km? How is the PSD 'smoothed'? The

page 5, line 9: it'd be good to specify the order of the Butterworth filter used.

page 5, line 16: 'more perturbed' means larger amplitude?

page 5, line 18: why this is due to propagating conditions, not the source?

page 7, line 2-5: It's not clear which panels in figures this sentence refers to. The wave activity difference is obvious at 50 km, but not at 60 km. The sentence needs to be more specific for the readers to make the comparison. Also, why using 6-14 hr for the vertically-filtered then using 8-24 hr for the temporally filtered? Use the same time range (e.g. 8-14 hr) for comparison makes more sense.

page 7, line 8: 'increased' to 'largest'

page 7, line 13-page 8 line 1: I think the short-term tidal variability is well recognized in the science community. It is not 'unexpected.' Even using satellite data, researchers have been trying to extract short-term variabilities, such as doi:10.1002/2016JA022528. page 8, line 10: The 'monthly mean' probably means 'average over the 14 days'?

page 8, line 10: 'relevant' to 'noticeable'

page 8, line 11: 'at an altitude of 50 km' to 'from 30 to 50 km'

page 8, line 23: besides -> aside from

page 8: line 26-27: This statement implies that the composite analysis does not include oscillations from GWs Doppler shifted into the tidal periods. I don't see how this can be the case. The wavelet method and the tidal fitting are no different. Neither can separate out the Doppler shifted GWs from tides.

page 8: line 5: 'that for' to 'from'

page 8, line 15-16: Where is the 'strong diurnal component' in Figure 6? I don't see any.

page 8, line 11-20: Similar to the discussion about Figure 4, the text often does not refer to specific panels in the figure, which makes it hard to understand what features the authors are pointing to.

page 10, line 11-12: Is this also due to sponge layer? If so it's better to state the sponge layer here than later on page 11.

page 12, last paragraph: While I agree with authors that ozone is perhaps not the main cause of the tidal variability during the 14 days, the argument here is not accurate. Because tides are global structure, they are forced and therefore related to the global structure of the ozone layer. Planetary scale perturbations of the ozone could cause tidal variability, but it may not show up as a correlation between local ozone concentration and local tidal amplitudes. Lack of correlation at a single site does not support the argument that they are not related because it could be a weaker ozone at other longitudes that causes the weaker global tides.

Figure 11: The amplitude of temperature perturbation is not a complete representation of GW energy. The potential energy, which is related to N squared is more appropriate. Because of the quick change in the temperature gradient from the stratosphere to the mesosphere around 50 km, the GW potential energy variation may be quite different from temperature perturbation amplitude.

---

## Author Comment (AC1) · 14 Sep 2017

**Author's response on "Temporal variability of tidal and gravity waves during a record long 10 day continuous lidar sounding" by Kathrin Baumgarten et al.**

**Anonymous Referee #1**

The paper by Kathrin Baumgarten and coauthors presents a continuous observation of middle atmosphere temperatures of unprecedented length. This is facilitated by newest advances in daylight capability as well as favorable observation conditions. The data are analyzed with a number of different techniques discussing the different effects of vertical and horizontal detrending. This is comprehensive and well presented. However, I do not agree with the interpretation of the findings in terms of a GW-induced change of the tidal amplitudes. This interpretation is based solely on the data without a clear discussion of the mechanisms. I would ask the authors to reconsider this part in the light of the major comments given below.

*We thank the reviewer for the valuable and constructive comments. We revised the manuscript with regard to your comments. Detailed answers are given below. The line numbers for the changes refer to the manuscript with marked changes.*

Major comments
The central figure in terms of interpretation is Figure 10. The figure presents the temporal variation of the tides, the GWs and the background winds. The tides show a general tendency of decrease over the 10-day period with some variation seen only in the temporally-filtered data. GWs show several peaks with the strongest peak after day 10, after the wind reversal of both zonal and meridional winds. That peak is more outstanding in the temporally filtered data, but amplitudes are much larger in the vertically filtered data.
The only point which supports your hypothesis is that the strongest decrease in tidal amplitude and the strongest peak of GW amplitudes appear both at the same day, i.e. day 11, in the temporally filtered data. However, this is a single event! It may be coincidence, the two variations could be caused by a common reason or it could indeed be causality. In order to argue that it is not a coincidence you would need more data. You claim that you can find a strong correlation: that could be tested by actually calculating correlation coefficients. (Proof me wrong, but I do not expect you to find a correlation significant based on Student's t-distribution.)
In order to argue for causality you would need to describe the mechanism in more detail. Here I see several difficulties.

*We agree that this event is only a single event from which we are not able to derive all details of the potential mechanism. However, our measurement shows a unique case for the variability of different waves on time scales of days in the atmosphere. In the revised manuscript, we make clear that our explanation of the observation is a hypothesis. Additionally to this, we mentioned other possibilities now in the text (P17L9).*

A) The (part-)global nature of tides. In general tides are understood to be a superposition of Eigenmodes of the atmosphere resonantly excited by solar heating (or chemical heating or latent heat release triggered by the solar cycle). In the superposition of different modes there may be locally varying amplitudes depending on the place of the

observation. The relative phases then also would decide on the amplitude at a single observing station. In addition, non migrating tides may be excited by the interaction of tides and planetary waves (cf. e.g. Liebermann et al., JGR-Atmos., 2015). By the way, it is not correct that satellites do not observe short term variability, see this paper and also e.g. Pedatella et al., JGR-space, 2016.

If you want to understand the cause for the variation at Kuehlungsborn you have to first find out which tidal modes are involved and whether these modes change amplitude or rather phase. Then you would need to look for mechanisms on a part-global scale. The column above Kuehlungsborn is definitely insufficient.

*Thank you for the comment. It is right tides in the atmosphere have a global nature. To completely understand their variation it would be necessary to use also data sets which cover them globally. But for a comparison to our ground-based measurements we need to measure the same tidal modes. These cannot be clearly identified from ground-based measurements. From our measurement we also calculated the mean phases of the tidal components over the three time intervals shown in Fig. 5 to investigate whether a possible change of tidal modes could have happened. The corresponding phase of the diurnal component shows no phase jump. Therefore, we assume there was no change in the tidal modes during the different time periods. This is not the case for the terdiurnal tidal signature. But the main focus in this paper is the variation of the diurnal tidal signature. Unfortunately, a ground-based measurement is limited in the observation of these tides, as it is, e.g., not sufficient to distinguish between migrating and non-migrating tides. This could give also potentially rise to a variation of the observed tides with the lidar. We mentioned these other possibilities in the manuscript now (P17L10).*

*The provided short-term variability of tides from satellite measurements by Lieberman et al., 2015 and Pedatella et al., 2016 are based on the calculations from Oberheide et al., 2002. This method has some limitations why we think this it is not suitable to access a day-to-day variability of the diurnal, semidiurnal and terdiurnal tides at the latitudes discussed here in our study.*

*The difficulties are:*
- *applies only to low latitudes and non migrating diurnal tides (Oberheide et al., 2002)*
- *PW activity is not taken into account (but is relevant at latitudes higher than 50°N)*
- *the LST separation between ascending and descending nodes decreases for latitudes above 50°N (Oberheide et al., 2002)*

*As a result our lidar observations present a unique case to illustrate the tidal variability on short time scales, which are not accessible by other remote sensing techniques so far. We added some of these information in the introduction (P2L22).*

B) You would still have a stronger argument if you could show that the event considered is exceptionally strong. A single event may dominate the momentum flux of a whole latitude circle. You have climatologies from you own observations as well as from satellite data. Put your event into a context! That is a value of its own. In your discussion at the moment you focus on the temporally filtered data, because there you have only on peak while you have three in the vertically filtered data. However, GW momentum flux is proportional to the square of the temperature amplitude, which is a factor 9 (amplitude squared) larger in the vertically filtered data. Also the latter would be more susceptible to wind filtering and dissipation.

*We have followed your recommendation and we have tried to put the event into a wider context. The diurnal tidal amplitude is stronger in May 2016 than the mean value of May from former years (P15L21). To compare the lidar measurement with further global tidal fields derived for instance from satellite measurements, we would need the same data retrieval and the same accuracy of data acquisition. But unfortunately, this is hard to achieve either because the altitude range covered is not the same or the temporal resolution is quite different.*

C) Also GWs propagate obliquely. And what causes the peaks? The question what the origin of the peaks in GW amplitude might be is probably the most obvious one. Again you try to answer this solely from the column above Kuehlungsborn. The wind reversal points to some cyclone or anticyclone in the vicinity. Is there frontal activity? Please consider also weather maps and search for potential sources. Also high resolution ECMWF data may be helpful (you show quite remarkable agreement) to find potential sources.

*It is absolutely right that an explanation solely from the column above Kühlungsborn is not possible. Our conclusion is more like a hypothesis or a speculation. Obviously also a change in the gravity wave source could have happened during the measurement period. As gravity waves can propagate hundred or thousand kilometers from their source region away, it needs further investigations to retrieve their origin using ray-tracing algorithms and additional model data. Referring to next comment of the referee also UWaDi could be helpful here, but UWaDi is only able to provide the absolute wave numbers without the sign and also 3D fields of wind and temperature from a model are needed. In our opinion this is out of the scope of this paper and needs to be done in a future study.*

Overall, my recommendation is to strengthen the observational part by putting the observed event into a context: is it an unusually strong event? In addition, dig somewhat deeper into the sources of the GWs with the help of NWP data, potentially considering high resolution ECMWF data. Could UWaDi be helpful? Keep the ozone discussion short, local ozone does not tell you much. Then at the end you may suggest that the GWs could have an influence on the tides but do not build your whole paper around this hypothesis and do not quote this as an "elegant demonstration of the strong impact of gravity waves on the diurnal tide": you are missing the evidence!

*We agree with your recommendation. Therefore, we have limited our interpretation and have mentioned also other possibilities. In the end, we cannot provide the evidence or the true explanation for our observation, but our study provides a unique case of the short-term variation of a diurnal oscillation, which could be related to a short periodic wave. The variation of the diurnal wave needs to be taken into account in the future when gravity wave parameters are retrieved from such ground-based measurements because there it is often assumed that the tidal background is constant over time (P17L9 and P18L16).*

Specific comments:

P1L2 infected -> affected *Done*

P1L22 Separate your references for GWs and tides. There is also tidal excitation due to latent heat release, i.e. convection. *Done*

P2L1 therefore -> accordingly *Done*

P2L10 ground based = what remains on the ground = radar, lidar, airglow radiosonde, rocket: in situ *Done*

P2L35 data during the 10 days of continuous lidar data in May 2016 and *Done*

P3L6 flash-lamp pumped *Corrected to flashlamp pumped.*

P3L7 better signal-to-noise? In which way?

*We are basically using Rayleigh scattering, which cross section is proportional to $\lambda^{-4}$. This means the scattering is much more effective for a wavelength of 532 nm compared to the fundamental wavelength of 1064 nm.*

P3L27 below the initial retrieval altitude ? *This is right, we clarified the sentence ending.*

P4L9 over the entire period are shown in Figure 1 as well. *Done*

P4L12 only contain tidal waves*;* also gravity waves as *Following the suggestion from referee 2 we have changed the sentence.*

P5L1 calculated using as a <- one or the other *Done*

P5L2 Please include the 50km altitude also in the text *Done*

P5L17 Within this altitude range clear waves What do you mean? The altitude range 40-55km or the ranges below and above? I cannot follow your discussion here: In both cases I see waves, though at different frequencies. Maybe waves in the upper panel weaken at the highest altitude. Is that the effect of the vertical Butterworth filter? It also appears to me that the change between receiving channels also causes a noncontinuity in the waves - though that is hard to say as the line is in.

*To clarify this we have added a statement in the manuscript with the altitude information (P5L14). The wave structures from the vertical filtered data are not as regular as from the temporally filtered data. This is not an effect of the vertical filtering. The vertical Butterworth filter could lead indeed to errors, but only for the last altitude bin what can be seen at 65 km. Another point you mentioned is a possible non-continuity in the structures because of a change of the receiving channels. The vertical filtering leads to a smoothing of this discontinuity, therefore it is unlikely that this is the reason for a change in the wave structures.*

P6F3 vertically-filtered data
Please show also the low-pass filtered data in Fig.3

*We have changed Fig. 3 accordingly. Now the temperature deviations induced by gravity waves and tides are shown. Also some text was added to describe Fig. 3 (P6L3).*

P7F4 Wavelet spectra *Done*

P8L1 The unexpected decrease of the diurnal component shown above indicates a strong short-term variability for tidal components What precisely do you want to infer here? That the tides have a boost at 50km which not yet reached 60km? Then: what is the vertical group velocity of tides? Your period is quite long. Shouldn't the high amplitudes you see at 6 May reach a few days later also 60km? Or do you mean simply that there is a strong variation at 50km with a peak at 6 May? Please clarify.

*We apologize for the misleading statement, the interesting point here was indeed the strong variation over time at 50 km. We clarified it now (P7L28).*

P8L1 referred to be constant (e.g. from the satellite community). a) Also in the satellite community there are quite a number of approaches b) Often some temporal coherency is used to identify a number of different migrating and non-migrating modes. These, evaluated at a single location, may result in apparently strongly varying amplitudes, albeit a constant amplitude was assumed for all these global modes. c) That amplitudes are assumed to be constant is then usually just the lack of data. So far we unfortunately do not possess the perfect observations with simultaneously global coverage and good coverage of local time.
Please omit the half sentence here. You can include some discussion in the introduction or summary section.

*We followed your suggestion and omit the half sentence. Additionally, we have included some information in the introduction regarding the satellite measurement (P2L22).*

P8L15 That you need a closer look tells you that the differences cannot be huge ... Are you even sure that they are significant? Probably yes, as they seem to be consistent over a wider altitude range. However, that they match at 43km does not seem to be significant to me given the variations in the profile above.

*The differences are significant as the first interval show clearly larger amplitudes for the diurnal variation which is indeed consistent over a wider altitude range. We changed the beginning of the sentence as it was misleading before (P9L4).*

P8L22 50 % : Sorry, I am bewildered. In Fig. 5 left panel, blue curves you have at 44km a mean amplitude of roughly 1.5K, and a minimum (dashed) of 1K = 30%. Just above at 46km the three curves are inside .2K at 2K mean amplitude, i.e. about 5% variation around the mean. For a global mode with long vertical wavelength such as a tide I would not assume the minor zigzagging to be real but indicative of the precision with which you can determine your amplitudes. On the other hand, the wavelet analysis indeed shows a change of a factor 2 in amplitude, so I am missing that special part of consistency. My interpretation would be that tidal amplitudes are stable, if periods of 7 days and larger are considered but that on shorter scales the local tidal amplitudes vary strongly.

*The minimum value of the diurnal tidal amplitude from the second period is generally between 30 and 50 % lower than the maximum value from the first period in the altitude range of 43 – 58 km (except of the values at 46 km and 55 km). We rephrased the sentence to make clear what values are compared with each other (P9L12).*

P8L26 which could be Doppler shifted to *intrinsic* periods larger than the Coriolis Period

*We have clarified in the manuscript which period we are referring to (P10L1).*

P8L27 In the composites you implicitly assume a constant phase of the tide over the analysis interval. Phase variations hence would also be a reason for different results.

*You are right. The basic assumption in the composite analysis is a constant tidal phase during the single days used for the composite. Furthermore we assume that the phases of GW are more randomly distributed. Due to the overlaying of the data according to the local solar time, GW are*

*averaged out and the remaining waves are probably tidal signatures. We added such a statement in the manuscript (P10L2).*

P9L17 we assume that the disappearance -> we want to investigate whether ????

*We rephrased the sentence.*

P10L2 depends on wind conditions as well as on their interaction. Please be more precise, e.g. the propagation conditions of tides depend on the mean background winds and the propagation of GWs both on the mean wind and the tides.

*We changed the sentence to be more precise (P11L2).*

P10L6 ECMWF is able to reproduce the meteorological situation above Kuehlungsborn. *Done*

P11L8 the sponge layer and the fact that there are basically no data above the stratopause assimilated.

*Aside from the sponge layer, that there is no data assimilation above the stratopause is also a reason why the data reliability becomes worse at larger altitudes. Now, this is also mentioned in the text (P11L13).*

P12L5 What do you mean: that the variation, albeit weak, is caused by PW or that the weakness of the perturbation is caused by PW

*Sorry for the misunderstanding, we meant the general variation is presumably caused by planetary waves. We clarified this in the text (P13L7).*

Fig10: Please assign panel indices (a,b,...). There seems to be a data gap after day 10 in the observations. There are some odd blue lines at the bottom of the plots in the middle row.

*We removed the odd blue lines in the plots and added some panel indices for readability. There are also two data gaps in the data (duration ~5 h and ~1 h). But both gaps are small enough to interpolate them easily and therefore they have no effect on the results (P4L18).*

---

## Author Comment (AC2)

**Author's response on "Temporal variability of tidal and gravity waves during a record long 10 day continuous lidar sounding" by Kathrin Baumgarten et al.**

**Anonymous Referee #2**

This paper presents a study of gravity wave and tidal activities in the stratosphere and lower mesosphere using lidar temperature measurements that span continuously over 14 days. Such continuous measurement is unprecedented and is extremely valuable for the study of wave propagation through this region and interaction of waves at different time scales. This work is fairly thorough, and the key points are well described and supported by analyses from various angles. Because of this, I think this work should be published. I do find some various places where the manuscript can be improved and have given my suggestions below.
Aside from that, it's important to note that tides are global features while GWs are highly localized. The decrease of diurnal oscillation seen in this measurement does not necessarily mean a decrease of the global diurnal tide amplitude. Such decrease could be due to interactions among different tidal modes or with planetary waves. Nevertheless, I think the analysis in this work does make a case that it is possible the local decrease of the diurnal oscillation is related to a GW. The authors do need to carefully distinguish between diurnal tides (global) and diurnal oscillation (local) in the text.

*We thank the reviewer for the helpful and constructive comments. The line numbers for the changes refer to the manuscript with marked changes. Regarding the global features of tides, we added a statement in the manuscript, which makes clear that the diurnal variations observed by the lidar are not necessarily features of the global tidal field (P6L3). Detailed answers to the comments are given below.*

page 2, line 25: This is perhaps not a fair comparison. A lidar's capability is not measured by the altitude range it can measure (unless they all measure the same thing). The Rayleigh and metal lidars measure different regions and serve different purposes. If the lidar data used in this study included the mesopause region, then this statement would be appropriate.

*Our statement was too general. A lidar using only one scattering mechanism is not enough to cover the entire middle atmosphere. We have rephrased the sentence to make clear, why especially the RMR lidar at Kühlungsborn is a better tool to analyze the short-term variability of atmospheric waves than other remote sensing techniques (e.g., radars) (P2L32).*

page 3, line 26: It is not clear how 'strong solar background' is related to the starting altitude. Does it make it lower or higher? Please explain.

*The signal-to-noise ratio depends on the solar background. It becomes worse during the day. The altitude of the start temperature should be a little bit above the altitude where the noise overcomes the signal level. Otherwise artificially added variations could be wrongly interpreted as wave variations.*

page 3, line 31-32: please specify the cut-off periods/vertical wavelength of these filters. *Done*

page 4, line 12: suggest changing to '... data set contains not only ..., but also ...' *Done*

page 5, line 1: remove 'using' *Done*

page 5, Figure 2: Since the data is averaged for 2 hrs (stated on page 3, line 28), the

highest frequency that can be resolved is 1/(2hr). Even though the data point is every 15 min, the figure better not extends to higher frequency because there is really no information beyond 1/(2 hr). Caption: not sure what the 'first' and 'second' half mean and how they relate to the error bars. Need clarification. It's not clear what data was used to calculate this spectrum. Is it from 50 km temperature only or average over an altitude range around 50 km? How is the PSD 'smoothed'? The

*We changed the plot to take into account that beyond a frequency of 1/(2 h) no real information is available. The data used for the plot are the temperature deviations from a mean temperature over the entire days (the 'unfiltered' data). To estimate an error of the spectrum, we have to acknowledge the variability over the time. As a first guess, we have calculated three different spectra at 50 km altitude. The difference here is related to the time which is covered for the calculation. Spectrum 1 is calculated from data over the whole time, the data for spectrum 2/3 covers the first/last 5 days, respectively. This is not entirely true for the actual error estimation, but it is sufficient to show a potential variation of the spectrum. The spectrum is smoothed with a Hanning filter (P4L28).*

page 5, line 9: it'd be good to specify the order of the Butterworth filter used. *Done*

page 5, line 16: 'more perturbed' means larger amplitude?

*In this case 'more perturbed' means the wave structures are less regular. This is independent from the amplitudes; it is only related to the occurring frequencies, which are much more different.*

page 5, line 18: why this is due to propagating conditions, not the source?

*It could also be related to different sources of gravity waves, from where the GW are able to propagate in this altitude region. We have added this in the manuscript (P6L1 and P15L31).*

page 7, line 2-5: It's not clear which panels in figures this sentence refers to. The wave activity difference is obvious at 50 km, but not at 60 km. The sentence needs to be more specific for the readers to make the comparison. Also, why using 6-14 hr for the vertically-filtered then using 8-24 hr for the temporally filtered? Use the same time range (e.g. 8-14 hr) for comparison makes more sense.

*We improved the text with additional references to the figure panels.*

page 7, line 8: 'increased' to 'largest'

*We decided to skip this change, because in this case, we only want to say that the wave amplitudes are grown compared to below. That the diurnal wave signature is largest here is another fact which is mentioned later.*

page 7, line 13-page 8 line 1: I think the short-term tidal variability is well recognized in the science community. It is not 'unexpected.' Even using satellite data, researchers have been trying to extract short-term variabilities, such as doi:10.1002/2016JA022528. page 8, line 10: The 'monthly mean' probably means 'average over the 14 days'?

*The monthly mean is an average over the 14 observations. The name indicates that the values are representative for the month May. We removed the word 'unexpected' to take into account that even satellite retrieval try to extract the short-term variability of tides. We added some information about these approaches in the introduction (P2L22). There are some limitations in the methods for the*

*extraction of tidal signals from satellites. We have mentioned these also in text as already written in the answers to referee 1.*

page 8, line 10: 'relevant' to 'noticeable' *Done*

page 8, line 11: 'at an altitude of 50 km' to 'from 30 to 50 km' *Done*

page 8, line 23: besides -> aside from *Done*

page 8: line 26-27: This statement implies that the composite analysis does not include oscillations from GWs Doppler shifted into the tidal periods. I don't see how this can be the case. The wavelet method and the tidal fitting are no different. Neither can separate out the Doppler shifted GWs from tides.

*During the composite analysis the data are sorted with respect to the local solar time based on the assumption that the tidal phase is constant over time compared to the phase of gravity waves. GW are assumed to have randomly distributed phases because of their different excitation mechanisms. All waves which are randomly produced (GW) will be averaged out in the composite. Therefore we assume that only tidal signatures are remaining in the data. Such a statement is now within the text (P10L2).*

page 8: line 5: 'that for' to 'from' *Done*

page 8, line 15-16: Where is the 'strong diurnal component' in Figure 6? I don't see any.

*In Figure 6 no strong tidal features can be seen. The expression was related to Figure 4 just to remind the reader to the tidal variability compared to the GW variability in Figure 6.*

page 8, line 11-20: Similar to the discussion about Figure 4, the text often does not refer to specific panels in the figure, which makes it hard to understand what features the authors are pointing to.

*We improved the text to clarify which Figure panel is described.*

page 10, line 11-12: Is this also due to sponge layer? If so it's better to state the sponge layer here than later on page 11.

*This is right. We mentioned this in the text now (P11L13).*

page 12, last paragraph: While I agree with authors that ozone is perhaps not the main cause of the tidal variability during the 14 days, the argument here is not accurate. Because tides are global structure, they are forced and therefore related to the global structure of the ozone layer. Planetary scale perturbations of the ozone could cause tidal variability, but it may not show up as a correlation between local ozone concentration and local tidal amplitudes. Lack of correlation at a single site does not support the argument that they are not related because it could be a weaker ozone at other longitudes that causes the weaker global tides.

*We mention the possibility of a change of ozone at other longitudes now in the text (P14L3).*

Figure 11: The amplitude of temperature perturbation is not a complete representation of GW energy. The potential energy, which is related to N squared is more appropriate.

Because of the quick change in the temperature gradient from the stratosphere to the mesosphere around 50 km, the GW potential energy variation may be quite different from temperature perturbation amplitude.

*We agree with your statement that temperature amplitudes do not completely represent the GW energy. But in this case, this is not necessary, because the purpose of Figure 11 is to demonstrate that the strongest decrease of the diurnal oscillation appear in the same altitude range as the GW from the temporally filtered data. An influence of a changing N is minimized due to the normalization and therefore it is negligible. Additionally to this, GW from the temporally filtered data have large vertical wavelengths, they are of a similar scale like the tidal wave signal. Consequently, they might be able to interact.*

---

## Author Response (AR2)

**Author's response on "Temporal variability of tidal and gravity waves during a record long 10 day continuous lidar sounding" by Kathrin Baumgarten et al.**

**Anonymous Referee #1**

The paper presents a time series of unprecidented length and altitude coverage. Gravity waves and tidal waves are simultaneously analysed from these data and interpreted. There are indications for GW tidal interactions. The major comments from the first submission are taken into account in the revised version. The paper hence should be published in ACP. However, there are some technical points regarding readibility. Nothing of this is difficult to correct, but since there is quite a number I recommend minor revision.

*We thank the reviewer again for the last minor comments. We have changed the manuscript with respect to the comments. Detailed answers are given below.*

Specific comments:

As a general advice: number all panels of your figures individually. References in the text become shorter and easier to read (e.g. Fig. 6a instead of Fig. 6 upper left panel)

*For a better readability we have changed the numbering of the panels in Fig. 4 and Fig. 6.*

P1L4 and not adequately implemented *Done*

P1L10/L13 during the measurement period *Done*

P1L22 through -> by ? structures, by convective ...*Done*

P2L14 Hertzog would be superpressure balloons which is not in your list. Maybe use here, in addition, one of the overview papers again (FA2003,A2010) *Done*

P2L15 a global *Done*

P3L13 results -> findings *Done*

P4L19 mean temperature profile averaged over *Done*

P4L23 data sets do *Done*

Fig 3: Better have a 2-column figure, reduced to one print column it is too small. Label all 4 panels a to d. *Done*

P6L-4 ... referred to as ... However, we ... mention that ...*Done*

P6L-2 ... not necessarily related to global tidal ...
Tides are (part-)global, also the non-migrating tides, otherwise you need to call them differently (e.g. 24h period GW). ... related to *single* global ... ?

*We would like to stick to our phrasing. a) There is general agreement to call oscillations with 8/12/24 h period as "tides", even if a final confirmation cannot be given from lidar data. b) Several recent observations and model studies suggest that tidal amplitudes and phases vary due to a "local" superposition of tidal modes and due to "local" filtering processes. Therefore, tides are not necessarily constant at a particular longitude.*

P7L13 sounding period *Done*

P7L17 or the most of the waves occuring in this period range *Done*

P7L19 is increased as well in the vertically ... as in the ... The vertically filtered data show ...*Done*

P7L23 between the 10th of May or between 10 May ...*Done*

P7L26 ?? which has to be acknowledged ?? What do you mean?

*We have added a half-sentence to make clear what was meant.*

P9L3 You mean due to the fact that you have a larger signal from your analysis, if the phase of the tide remains locked during the analysis period?
Perhaps: due to phase variations over the particular month analyzed
(Instrument noise should not influence the outcome, if you have a sufficient number of independent data)

*The signal-to-noise ratio in the upper range of the lidar sounding depends on the weather conditions. For every particular month we had differences above 65 km. Therefore, the variance in higher altitudes is larger compared to lower altitudes, which overlays the original tidal variation. Phase variations during a particular analyzed month typically reduce the average amplitude as demonstrated in Fig. 5. Vice versa, if there is no phase variation within the analyzed lidar data, the amplitude may appear higher compared to a complete month.*

P9L4 ... intervals (dashed and dotted lines) ... reveals large ...
(Order 30%; for "huge" I would expect an order of magnitude at least) *Done*

P9L10 On the contrary omit, there is no contradiction *Done*

P9L14 restricted to tides*;* also other *Done*

P9L15 That's a bit short, perhaps better:
... included in the data. For instance, amplitudes may include some gravity waves which have observed ground-based periods larger than the Coriolis period but are Doppler shifted to intrinsic frequencies in the range of gravity waves.

*We clarified the statement and changed the sentence as follows "For instance, amplitudes may include some long-period gravity waves which are Doppler shifted to observed periods larger than the Coriolis period, i.e. to periods in the range of 24 h."*

P9L17 You are quite frequently first giving the result and explaining it afterwards. O.k., if you have a complicated explanation, but disrupts the flow for shorter arguments. Better switch the order of the sentences:

... shifted to intrinsic frequencies in the range of gravity waves. The composite analysis ... Therefore,

gravity waves average ... The occurency of low frequency gravity waves hence leads to higher amplitudes in the ...

*We rearranged the sentence as proposed.*

P10L4 For comparison, ...*Done*

P11L16 reveal -> investigate *Done*

P11L20 In general, ECMWF data exhibit similar structures than the lidar data shown in Fig. 1. Temperature deviations ... *Done*

P11L23/L26 However, ...*Done*

P11L28 That sentence is a conclusion not a contradiction Hence, ... Accordingly, ...
... information of the background atmosphere and to a lessser degree also for the wave fields ...*Done*

P11L31 Therefore, should be something like To this purpose, ... , but better omit *Done*

P12L13 ... planetary wave activity in the stratosphere is ...*Done*

P13L10 reliable -> likely *Done*

P13L11 a relevant *Done*

P14L3 for several consecutive days ... an exceptionally long *Done*

P14L8 looking at *Done*

P14L10 of different types of atmospheric *Done*

P14L11 Between altitudes ...*Done*

P14L11 shows an increase as expected due to the decreasing air density
I was wondering that you did not comment on that earlier. Between 35 and 47km, i.e. over less than two scale heights you find an increase of more than a factor of 4. Since the conservative quantities are proportional to the amplitudes squared, they should hence increase by less than a factor e= 2.7, so actually amplitudes increase more strongly than a conservative propagation would let expect.

*We want to avoid a quantitative description here, because this cannot be done based on temperature lidar data only. There are various reasons for deviations from a theoretically expected conservative propagation of the waves, e.g., waves entering or leaving the lidar field of view and/or conversion of kinetic into potential energy. We have slightly changed our sentence to weaken our description.*

P14L12 The not visible statement is only true for wavelet, though you have some indication also for the composite analysis. Please be more precise.

*We especially want to emphasize the variation with time, while the variation with amplitude below the stratopause is less surprising. Please compare to the previous comment.*

P15L11 estimation -> assumption illustrate -> exhibit / show *Done*

P16L1 In this study, later on the amplitude is indeed *Done*

P16L10 hypothesis / mechanism *Done*

P16L11 In which time? "these" refers to? Please reformulate the sentence. And swap sentences with the result as the last sentence (cf. above) *Done*

P16L15 correlated -> occurs at the same time is related to *Done*

P16L16 estimate -> propose *Done*

P16L22 The way the sentence is written I was looking for four curves. Please reorder and reformulate e.g.:
... temporally filtered data are investigated. Figure 11 shows GWs for ... and tides briefly afterwards. These are the times when ...

*Sorry for the misleading sentence. We rephrased this as proposed.*

P18L13 Are you concretely planning to do this? Otherwise maybe reformulate:
In order to investigate this further more sophisticated model studies are required which can use our observations as a bench mark test.

*We reformulate the sentence accordingly.*

[revised manuscript text omitted]
 as well in the vertically filtered data as  in the temporally filtered data. The  vertically filtered data show amplitudes of 4 K for this component with the strongest occurrence on  6 - 7 May 2016. Later this component becomes weaker. This behavior is even more pronounced in the temporally filtered data, where amplitudes of up to 6 K arise for the diurnal component in the first days. The amplitudes decrease to less than 1.5 K between  10 May and the end of the measurement period. Other components with periods between 8 and 12 - 13 h are also visible, but they reveal smaller amplitudes and are less persistent compared to the diurnal component. The decrease over time of the diurnal component shown above indicates a strong short-term variability for tidal components, which has to be acknowledged for the extraction of gravity waves. At an altitude of 60 km ( shown in Fig. 4 a and b) this intermittency of the tidal signature becomes

[revised manuscript text omitted]